# Polyphenols-Enrichment of Vienna Sausages Using Microcapsules Containing Acidic Aqueous Extract of *Boletus edulis* Mushrooms

**DOI:** 10.3390/foods13070979

**Published:** 2024-03-22

**Authors:** Melinda Fogarasi, Maria Jenica Urs, Maria-Ioana Socaciu, Floricuța Ranga, Cristina Anamaria Semeniuc, Dan Cristian Vodnar, Vlad Mureșan, Dorin Țibulcă, Szabolcs Fogarasi, Carmen Socaciu

**Affiliations:** 1Department of Food Engineering, University of Agricultural Sciences and Veterinary Medicine of Cluj-Napoca, 3-5 Mănăştur St., 400372 Cluj-Napoca, Romania; melinda.fogarasi@usamvcluj.ro (M.F.); ursmaria14@gmail.com (M.J.U.); maria-ioana.socaciu@usamvcluj.ro (M.-I.S.); vlad.muresan@usamvcluj.ro (V.M.); dorin.tibulca@usamvcluj.ro (D.Ț.); 2Department of Food Science, University of Agricultural Sciences and Veterinary Medicine of Cluj-Napoca, 3-5 Mănăştur St., 400372 Cluj-Napoca, Romania; floricutza_ro@yahoo.com (F.R.); dan.vodnar@usamvcluj.ro (D.C.V.); carmen.socaciu@usamvcluj.ro (C.S.); 3Faculty of Chemistry and Chemical Engineering, Babeş-Bolyai University, 11 Arany Janos Str., 400028 Cluj-Napoca, Romania; szabolcs.fogarasi@ubbcluj.ro

**Keywords:** Vienna sausages, enrichment, polyphenols, microcapsules, extract, mushrooms

## Abstract

Polyphenols are ubiquitous by-products in many plant foods. Their intake has been linked to health benefits like the reduced incidence of cardiovascular disease, diabetes, and cancer. These bioactive compounds can be successfully extracted from *Boletus edulis* mushrooms with acidic water. However, such extract could influence the sensory or textural properties of the product to be enriched; this inconvenience can be avoided by microencapsulating it using spray drying. In this study, the Vienna sausages were reformulated by replacing 2% of the cured meat with microcapsules containing an acidic aqueous extract of *Boletus edulis* mushrooms and by replacing ice flakes, an ingredient that represents 22.9% of the manufacturing recipe, with ice cubes from the same extract aiming to obtain a polyphenol enriched product. The results showed a higher content of polyphenols in sausages with extract (VSe; 568.92 μg/g) and microcapsules (VSm; 523.03 μg/g) than in the control ones (455.41 μg/g), with significant differences for 2,4-dihydroxybenzoic acid, protocatechuic acid, and 1-*O*-galloyl-*β*-D-glucose. However, because of the oxidative stress caused to the microcapsules by the extract’s spray drying, VSm had the highest oxidation state. PV and TBARS levels varied with storage time in all formulations, but given the short period tested, they were well below the allowed/recommended limit. The extract, as such, negatively affected the appearance, odor, and taste of Vienna sausages. The microcapsules, instead, determined an increase in their acceptance rate among consumers; they also prevented moisture loss and color changes during storage. In conclusion, microcapsules are more suitable for use as a polyphenol enrichment ingredient in Vienna sausages than the extract.

## 1. Introduction

Vienna sausages (or wiener) belong to the cooked and smoked products class next to frankfurters (hot dogs), Bologna sausages, and knockwurst. They are made from emulsified (finely ground) meats and, according to FSIS regulation §319.180 [1], may contain a combination of fat and added water of up to 40% in the finished product, but no more than 30% fat [2]. Due to their nutritional value (especially protein and fat), ease of digestion, and convenience (ready-to-eat food), Vienna sausages are a staple for breakfast, lunch, or dinner [3]. However, they are not a source of vitamins, enzymes, or other bioactive compounds.

Enrichment and fortification of animal products were investigated for about six decades, but only some studies have focused on Vienna sausages. Srikanchai et al. [4], Khiaosa-ard et al. [5], and Komprda et al. [6] have enriched the Vienna sausages with *n*-3 fatty acids by feeding pigs tuna oil and fish oil. Vivar-Vera et al. [7] have investigated starfruit dietary fiber concentrate as a novel ingredient in Vienna sausages, while Peshuk et al. [8] explored the efficiency of Vienna sausages enriched with leucine on women with prescarcopenia in the postmenopausal period. More recently, Kang et al. [9] have tested the substitution of chicken breast with soybean protein emulsion on the physicochemical properties of Vienna sausage.

Polyphenols have antioxidant properties and are well known for their health benefits. *Boletus edulis* mushrooms are rich in polyphenolic compounds. A previous study optimized their extraction procedure; the water/acetic acid mixture (9:1, *v*/*v*) recovered the highest quantity of polyphenolic compounds from their powder [10]. However, no research has used such an extract to enrich Vienna sausages with polyphenols. Therefore, this work aimed to evaluate the effect of different strategies for using an acidic aqueous extract obtained by macerating *Boletus edulis* mushroom powder (either as such or microencapsulated) as an ingredient rich in polyphenols (that replaces pork meat) on the physicochemical, sensory, and textural properties of Vienna sausages, respectively, on their storage stability.

## 2. Materials and Methods

Three formulations of Vienna sausages were prepared [control (VSc), enriched with 1.5% microcapsules containing acidic aqueous extract of *Boletus edulis* mushrooms (VSm), and enriched with acidic aqueous extract of *Boletus edulis* mushrooms (VSe)] that were stored for 8 days at 4 °C. The three formulations were analyzed initially and on days 3, 5, and 8 of storage to determine the effects of enrichment and storage time on proximate composition (the content of moisture, protein, fat, ash, and total carbohydrate), energy value, freshness [pH and easily hydrolyzable nitrogen (EHN) content], color, total phenolic content (TPC), and oxidative status [peroxide value (PV) and thiobarbituric acid reactive substances (TBARS) content] of Vienna sausages. An HPLC method was used to quantify the individual phenolic compounds in Vienna sausage formulations, and sensory analysis tests were performed to determine the consumers’ preference, acceptability, and purchase intention.

### 2.1. Preparation of Mushroom Powder

Wild mushrooms of the *Boletus edulis* species, purchased from S.C. Alisa Funghi S.R.L. (Chichişa, Romania), were manually sorted according to size and appearance, cleaned of dirt and physical impurities, washed, drained, sliced, and dried for 24 h at 45 °C with a dehydrator (DEH-450; Biovita S.R.L., Cluj-Napoca, Romania). Next, they were ground, and the resulting powder was stored in amber glass jars in a cool, dry place until use.

Preparation of acidic aqueous extract of *Boletus edulis* mushrooms. It consisted of the maceration of *Boletus edulis* mushroom powder with acidic water (10% (*v*/*v*) acetic acid solution), as described by Fogarasi et al. [10], including the following steps: homogenization of mushroom powder with acidic water, in a ratio of 1:9 (*w*/*v*), at 21,500 rpm for 30 s using a laboratory homogenizer (T 18 digital Ultra-Turrax; IKA-Werke GmbH & Co. KG, Staufen, Germany), followed by orbital shaking (3005; GFL Gesellschaft für Labortechnik mbH, Burgwedel, Germany) of this mixture for 24 h at 150 rpm and centrifugation (Universal 320 R; Andreas Hettich GmbH & Co. KG, Tuttlingen, Germany) at 8981× *g* (9000 rpm) for 10 min at 4 °C, then by vacuum filtration of the supernatant. Part of this acidic aqueous extract of *Boletus edulis* mushrooms was frozen in cubes, and part was used to obtain the microcapsules. The acidic aqueous extract of *Boletus edulis* mushrooms was prepared in two replicates, one for each Vienna sausages batch.

### 2.2. Preparation of Monolayered Microcapsules Containing Acidic Aqueous Extract of Boletus edulis Mushrooms

The acidic aqueous extract of *Boletus edulis* mushrooms was mixed with microcrystalline cellulose (Merck KGaA, Darmstadt, Germany) at a 2% ratio (*w*/*w*) on a magnetic stirrer (RSLAB-11C; RSLab, Heraklion, Greece). Then, 1 L of this solution was dried using a mini spray dryer (B-290; BÜCHI Labortechnik AG, Flawil, Switzerland) with a single nozzle feed. The spraying nozzle diameter was 0.7 mm, and spraying parameters were the following: peristaltic pump speed of 10 mL/min, the inlet temperature of 126 °C, the outlet temperature of 82 °C, and 6 bars of air pressure that was fed at a flow of 1374 L/h during drying with 100% set fan speed. The dried powder was then collected in sealable bags at room temperature. The monolayered microcapsules containing acidic aqueous extract of *Boletus edulis* mushrooms were prepared in two replicates, one for each Vienna sausages batch.

### 2.3. Preparation of Vienna Sausages

Nine kilograms of pork chunks (purchased from Cina Carmangerie, Cluj-Napoca, Romania) were ground using a wolf machine (Bizerba SE & Co. KG, Balingen, Germany) through a 3 mm sieve, mixed with 0.18 kg non-iodized salt (Salrom, Bucharest, Romania), and then kept in a cold room for 72 h at 4 °C for curing. The cured meat was then divided into three equal portions (3 kg), which were used to prepare three formulations of Vienna sausages (control—VSc, enriched with microcapsules—VSm, and enriched with acidic aqueous extract of *Boletus edulis* mushrooms—VSe) according to the manufacturing recipes in Table 1. Two batches of Vienna sausages were prepared this way.

For each formulation, the cured meat was chopped, mixed with the other ingredients, and emulsified in the bowl of a meat cutter (Meprotec GmbH, Pasching, Austria). The batter thus obtained was stuffed in 20 mm diameter collagen casings (S.C. Darimex International S.R.L., Otopeni, Romania) using a vacuum filling machine (Düker-REX Fleischereimaschinen GmbH, Laufach, Germany). The Vienna sausages were manually portioned and linked (at 12–15 cm intervals) by twisting, then hung on racks.

The racks with Vienna sausages were placed into a smoking and scalding chamber (H. Maurer & Söhne Rauch- und Wärmetechnik GmbH & Co. KG, Reichenau, Germany), where they were subjected to the following technological operations: air drying for 30 min at 75 °C, smoking for 10 min at 75 °C, cooking for 20 min at 75 °C (time required to reach the internal (geometric center of the sausage) temperature of 70 °C), and shower cooling for 5 min. Next, the Vienna sausages were stored unpacked for 8 days at 4 °C to determine whether the extract, as such or microencapsulated, confers oxidative stability. Samples were taken at intervals of 2 days to evaluate the quality changes of the three Vienna sausage formulations during storage.

### 2.4. Analysis Methods

The methods published in ISO 1442:1997 [11], ISO 937:2023 [12], ISO 1443:1973 [13], and ISO 936:1998 [14] standards were used to determine the moisture, protein, fat, and ash content of Vienna sausages. The formula for calculating the total carbohydrate content (%) was 100 − (% moisture + % protein +% fat + % ash) [15], and the one for energy value (kcal/100 g) was 4 × (g protein + g carbohydrate) + 9 × (g fat) [16].

The pH was determined in the aqueous extract of Vienna sausages, prepared as described by Socaciu et al. [17], using a digital multi-parameter meter (InoLab Multi 9310 IDS; WTW, Weilheim, Germany). The easily hydrolyzable nitrogen (EHN) in Vienna sausages was quantified using the method from SR 9065-7:2007 [18] standard. These analyses were carried out in triplicate for all samples of each Vienna sausages batch.

The texture analysis involved the compression of the Vienna sausages sample (120 mm length × 20 mm diameter) using the TA7 acrylic blade attached to a 10 kg compression cell. It was carried out in a single cycle at a depth of 23 mm from the sample’s surface, with a speed of 1 mm/s. The CT3 texture analyzer (Brookfield Engineering Laboratories Inc., Middleboro, MA, USA) recorded load [N] as a function of time [s], with the parameter monitored in this analysis being hardness (maximum load of the compression cycle). The measurements were performed in triplicate for all samples of each Vienna sausages batch.

The color measurement of Vienna sausages was carried out with the NH300 colorimeter (3NH, Shenzhen, China; illuminant: D65; measuring aperture: Φ 8 mm; standard observer: 10°) based on the CIE *L***a***b** color system, as Socaciu et al. [19] detailed. It was performed in duplicate on six slices (1 cm thick) for all samples of each Vienna sausages batch.

Sensory analysis of Vienna sausages and determination of consumers’ acceptance and purchase intention was carried out using the tests proposed by Semeniuc et al. [16]. The 9-point hedonic scale test was used to evaluate the appearance, color, odor, taste, texture, and overall acceptability of Vienna sausages. In total, 82 panelists (65 women and 17 men aged 18–52) evaluated the Vienna sausages’ appearance, color, odor, taste, texture, and overall acceptability. Panelists were informed about the ingredients of Vienna sausages and asked to sign an informed consent form. They were instructed not to consume (drink or eat) anything or smoke for at least one hour before the evaluation session, which was held in separate booths. The acceptance rate of each Vienna sausage formulation was calculated using the following Formula (1):(1)$$Acceptance\;rate\;\left(\%\right)=\frac{X}{N}\times 100$$
where *X* is the formulation’s mean sensory score, and *N* is the maximum sensory score it received. An acceptance rate equal to or greater than 70% is considered reasonable [19].

Methanolic extracts were prepared for each formulation to determine TPC and individual polyphenolic compounds in Vienna sausages. To this, 5 g of sample (minced Vienna sausages) were mixed with 5 mL methanol–water solution (70%, *v*/*v*) for 1 min using a vortex (6776; Corning Life Sciences, Monterrey, Mexico), sonicated for 30 min in an ultrasonic bath (USC 300 THD; Singapore), centrifugated at 8981× *g* (9000 rpm) for 10 min at 4 °C (Universal 320 R; Andreas Hettich GmbH & Co. KG, Tuttlingen, Germany), and the supernatant was vacuum filtrated. Each extraction was performed in triplicate for all samples of each Vienna sausages batch. All extracts were kept at −18 °C until use. The TPC was determined using a double-beam UV–Vis spectrophotometer (UV-1900i; Shimadzu Scientific Instruments, Inc., Columbia, MD, USA) following the method described by Michiu et al. [20]. Individual polyphenolic compounds were quantified using the HPLC-DAD-ESI-MS method of Fogarasi et al. [10] on a liquid chromatography system (1200 HPLC; Agilent Technologies Inc., Palo Alto, CA, USA). Before chromatographic analysis, the extracts were filtered through polyamide syringes (0.45 μm pore size, 25 mm diameter).

The determination of PV in Vienna sausages was carried out as Semeniuc et al. [21] described, using the fat extracted with a mixture of chloroform–methanol (2:1, *v*/*v*). The TBARS were quantified according to the method of Socaciu et al. [17]. These analyses were performed in triplicate for all samples of each Vienna sausages batch.

### 2.5. Statistical Analysis

Minitab statistical software (version 19.1.1; LEAD Technologies, Inc., Charlotte, NC, USA) was used for data analysis. A one-way ANOVA analysis with post hoc Tukey’s test at a 95% confidence level (*p* < 0.05) was performed to determine the effect of enrichment and storage time on the characteristics of Vienna sausages. In addition, the relationship strength between pairs of parameters was measured by calculating Pearson’s correlation coefficient (*r*).

## 3. Results and Discussion

### 3.1. Physicochemical Properties of Vienna Sausages

The results of proximate analysis, energy value calculation, as well as determination of pH and EHN content in Vienna sausage formulations can be seen in Table 2. Replacing 2% of the cured meat mass with microcapsules containing acidic aqueous extract of *Boletus edulis* mushrooms has resulted in a Vienna sausages formulation enriched with 1.5% microcapsules (VSm; see Table 1), with a significantly lower moisture content (62.97%) from that of control Vienna sausages (VSc, 64.40%). The formulation prepared by substituting ice flakes, a 22.9% ingredient in the Vienna sausages recipe, with ice cubes from *Boletus edulis* acidic aqueous extract (VSe) has shown an even lower moisture content (59.84%); a possible explanation may be that the extract contributes to the total solids in VSe by containing solubilized substances in the acidic water used to macerate the mushroom powder. In support of our findings, Torres-Martínez et al. [22] found the highest soluble solids concentration in aqueous mushroom extract when they tested the effect of different solvents (water, ethanol, and a mixture of water–ethanol) on the physicochemical properties of edible mushroom extracts. Soluble solids include dissolved sugars, acids, and—at a trace level—vitamins, fructans, proteins, pigments, phenolics, and minerals [23]. These findings are consistent with earlier research. Kanwal et al. [24] noted that fortifying fish sausages with flaxseed oil microcapsules decreased the moisture content of the finished product. In the study by Stangierski et al. [25], the poultry sausages formulated with microencapsulated fish oil had a lower moisture content due to the extra dry matter from the microencapsulation material.

The ash content of the three Vienna sausage formulations was not significantly different: 1.44% in VSc, 1.50% in VSm, and 1.61% in VSe. These findings are consistent with those of Domínguez et al. [26], who reported no significant differences in the ash content of Frankfurter sausages with and without microencapsulated fish oil.

The fat content differed the most between formulations (9.84% in VSc, 7.08% in VSm, and 15.28% in VSe), but this was not due to the additions used (microcapsules or acidic aqueous extract of *Boletus edulis* mushrooms) since the mushrooms powder had a low lipid content; most likely, it was due to the non-homogeneous composition of pork meat used as raw material. Pork meat contains muscle, connective, adipose, vascular, and nervous tissues [27]; their proportions vary with the slaughtered individual’s fattening state [28]. Some meat cuts may have more fat, and the fat deposition between muscle fibers and bundles (fascicles) is not uniform [29]. The pork chunks used as raw material to prepare the Vienna sausage may originate from more porcine animals, hence the compositional differences between the Vienna sausage formulations.

As for the protein content, it was significantly higher in Vienna sausages formulated with microcapsules (16.28% in VSm) and aqueous extract (18.84% in VSe) than in the control sausages (15.02% in VSc) because the *Boletus edulis* mushrooms powder (from the composition of microcapsules and extract) was rich in protein (26.5%). The total carbohydrate content also significantly varied between formulations, being lower in VSe (4.45%) than in VSc (9.29%) and VSm (12.11%); it is due to the significant differences in the three formulations’ moisture, protein, and fat content (used in calculating this parameter). Its higher content in VSm than in VSc is determined by the carbohydrate surplus provided by the microcapsules (from the mushroom extract and cellulose used for their preparation), like in the study by Domínguez et al. [26].

Because of differences in the three formulations’ proximate composition, the energy value of VSe (231 kcal/100 g) was significantly higher than that of VSc (186 kcal/100 g) and of VSm (177 kcal/100 g); however, between VSc and VSm, there was no statistically significant difference in terms of energy value. Enrichment of Vienna sausages with microcapsules and acidic aqueous extract of *Boletus edulis* mushrooms also had a significant effect on their pH, decreasing from 6.40 in VSc to 6.27 in VSm and 4.94 in VSe (giving them a sour taste).

The moisture content significantly varied with storage time in VSc (see Table 2), decreasing from 64.40% (initially) to 62.14% (on the eighth storage day). In VSm, it did not change substantially during storage, being 62.97% initially and 61.71% on the final storage day; a possible explanation is that cellulose in the microcapsules provided stability to the Vienna sausages against water loss during storage, as in the work of Stangierski et al. [25]. In VSe, however, a significant difference was found between the initial moisture content (59.84%) and that on the eighth storage day (58.90%).

The protein content did not significantly change during storage in VSm (15.93–16.28%) and VSe (18.45–19.26%). However, it varied with storage time in VSc, significantly increasing from 15.02% initially to 16.47% on the eighth storage day due to its concentrating [a higher water loss in VSc (2.26%) during storage than in VSe (0.94%) or VSm (1.26%)].

The fat content ranged between 7.23 and 9.84% in VSc (with a significant difference between the initial day (9.84%) and the third storage day (7.23%) but non-significant with the other storage days) and between 6.79 and 10.60% in VSm (with an upward trend during storage); in VSe, it did not significantly vary with storage time, framing between 14.73 and 15.50%. An explanation for the large difference in fat content between VSe and the other two formulations, not justified by the aqueous extract usage (which contains an insignificant amount of lipid) to their preparation, is the inhomogeneity of its distribution at the cellular level in pork meat. As for total carbohydrate content during storage, an in-the-mirror behavior with the fat content can be noticed, varying between 9.29 and 11.82% in VSc, 10.06 and 13.46% in VSm, and 4.15 and 5.90% in VSe.

The ash content significantly varied with storage time in VSe (between 1.36% on the third storage day and 1.80% on the fifth) but not significant in VSc (1.37–1.53%) or VSm (1.30–1.55%), which is also explained by the raw material’s inhomogeneity.

All these variations in the proximate composition of Vienna sausage formulations during storage have also led to significant changes in terms of their energy value, ranging between 178 and 190 kcal/100 g in VSc, 177 and 200 kcal/100 g in VSm, while in VSe, between 231 and 235 kcal/100 g. The slightly higher energy values from the end of the storage period are mainly due to the water losses in Vienna sausage formulations; as the difference between the initial energy value of VSm and the final one was lower than at the other two formulations, it suggests that this one was more stable at storage from this viewpoint. As in our case, Pourashouri et al. [30] have not found significant differences in the proximate composition of fish sausages fortified with fish oil, oil-in-water emulsion, gelled oil-in-water emulsion, and lyophilized aqueous extract of green tea during storage.

The pH did not significantly vary with storage time in VSc (6.37–6.41) and VSe (4.94–5.00). However, in VSm, it increased from an initial value of 6.26 to 6.36 (on the third storage day) and then to 6.39 (on the eighth day); this was probably due to the spoilage microorganisms’ activity, which resulted in the production of ammonia, amines, and other alkaline substances [31]. Unlike us, Stangierski et al. [25] reported that sausages with microcapsules had the lowest pH in the last two storage times (day 14 and day 21).

The EHN content in VSc significantly increased up to the fifth storage day (20.19 mg NH_3_/100 g) from 13.60 mg NH_3_/100 g initially and then decreased to 15.27 mg NH_3_/100 g on the eighth day [non-significant content compared to that on the third day (16.03 mg NH_3_/100 g)]. A possible explanation for this trend is the accumulation of acidic substances in sausages during this storage interval (days 5–8) that neutralized the ammonia, with the lack of a significant change in the pH with storage time pointing in the same direction. The Pearson correlation analysis indicates a moderate negative correlation (*r* = 0.54; *p* = 0.037) between the pH and EHN content in this formulation during storage; this shows that besides ammonia other alkaline compounds can occur in sausages during storage.

In VSm, the EHN significantly increased from an initial content of 18.97 to 22.73 mg NH_3_/100 g on the third storage day, after which it varied non-significantly, reaching 23.75 mg NH_3_/100 g on the fifth day and 22.10 mg NH_3_/100 g on the last storage day. A moderate positive correlation between EHN content and pH was found for this formulation (*r* = 0.62; *p* = 0.021), which means that, as the ammonia accumulated in Vienna sausages during storage, they became less acidic.

As for VSe, the data revealed a zig-zagging pattern of EHN content with storage time: an increase from 18.53 to 21.10 mg NH_3_/100 g (on day 3 of storage), then a decrease to 18.68 mg NH_3_/100 g (on day 5 of storage), and again an increase to 22.27 mg NH_3_/100 g (on day 8 of storage). No significant relationship (*r* = 0.09; *p* = 0.480) was found between EHN content and pH for this formulation.

The level of EHN was, at all storage times, lower in VSc compared to VSm and VSe, probably because it was the first formulation processed that day among the three. However, the increasing rate of EHN content was higher in VSc (by an average of 1.3) compared to VSm (by 1.2) and VSe (by 1.1). The maximum permissible level for the EHN content in Vienna sausages is 30 mg NH_3_/100 g [32], which was not exceeded for any formulation during the eight days of storage.

### 3.2. Colour Properties of Vienna Sausages

The measured values for color attributes of Vienna sausage formulations are shown in Table 3. The use of microcapsules, in a concentration of 1.5% in the finished product, to formulate VSm has caused a significant intensification of the lightness of sausages and yellow shade but a fading of the red shade; all this because the microcapsules had a white color with a slight yellowish shade. These results confirm those of Cha et al. [33]; in their study, the pork patties’ lightness and yellowness increased when formulated by partially substituting meat with white jelly mushrooms.

However, the lightness faded, and the shades of red and yellow significantly intensified in VSe because the acidic aqueous extract of *Boletus edulis* mushrooms used as an ingredient (22.9%) was yellow-reddish. Given that the hue angle had values close to 1 in all formulations, they can be characterized as red; as for the color intensity of Vienna sausages, the highest Chorma value was found in VSe, followed by VSm and VSc. All these changes in color attributes inflicted by the usage of microcapsules or acidic aqueous extract of *Boletus edulis* mushrooms in the preparation of Vienna sausages have led to a total color difference (Δ*E*) of 2.07 (noticeable) between VSm and VSc and 9.29 (large) between VSe and VSc.

The *L**-value significantly varied with storage time in VSc and VSe (see Table 3) but non-significantly in VSm (between 71.80 and 72.43). In VSc, it significantly decreased from an initial value of 70.86 to a final one of 68.82, and, in VSe, from 64.73 to 61.57, with a non-significant variation up to the third storage day.

The *a**-value significantly varied in all three formulations during storage. The difference was non-significant between the initial value (10.29) and that on the eighth storage day (10.25) in VSc, but it was significantly higher on the eighth storage day (9.95) than initially (9.68) in VSe; this means an intensification of the red shade in VSe during storage, probably due to concentration by dehydration [17]. Instead, the *a**-value was significantly lower on the eighth storage day (16.04) than initially (16.69) in VSe, revealing a fading of their red shade with storage time, most likely because of oxidation of heme pigments [34].

The *b**-value significantly varied with storage time only in VSm, being higher on the eighth storage day (19.21) than initially (18.93); this shows an intensification of the yellow shade in this formulation at storage, probably caused by fat oxidation [17]. This color attribute ranged between 17.35 and 17.71 in VSm and 19.70 and 21.48 in VSe. In Wan Rosli et al.’s [35] study, adding oyster mushrooms decreased the lightness and yellowness of chicken patties without affecting their redness.

The hue angle, which shows the dominant color shade of Vienna sausages (namely red), underwent non-significant variations with storage time in VSm (1.09–1.10) but significant in VSc (1.02–1.05) and VSe (0.87–0.93); the Croma index, which provides information about the color intensity of Vienna sausages, showed non-significant variations during storage in VSc (20.18–20.51) but significant in VSm (21.27–22.02) and VSe (25.39–26.81). These variations were explained above in the discussion of *a**-value.

The Δ*E** between VSm and VSc ranged during storage from 2.07 to 3.55 (from noticeable to appreciable) and between VSc and VSe from 8.11 to 10.23 (which means large).

### 3.3. Sensory and Texture Properties of Vienna Sausages

VSe scored significantly lower than VSc and VSm for all sensory attributes evaluated (see Table 4); the acidic mushroom extract destabilized the meat emulsion (so-called meat batter) when mixed in the cutter’s bowl with the other ingredients, giving the Vienna sausages an unpleasant appearance, pungent odor, and sour taste. The overall score was calculated by averaging the scores given to a formulation for appearance, color, odor, taste, texture, and overall acceptability; VSm had the highest overall score (7.8 points), followed by VSc (7.8 points), although there was no significant difference between these. VSe had a significantly lower overall score (4.8 points) than the other two formulations, mainly due to their appearance, color, and taste. The texture of VSe also received a low sensory score (4.9 points); however, there were no significant differences between the hardness values (instrumentally determined) of the three formulations (41.12 ± 0.348 N for VSc, 42.66 ± 2.205 N for VSm, and 43.58 ± 1.142 N for VSe). In the study from Wang et al. [36], where the shiitake mushrooms were used as the meat replacer in the production of sausages, the sensory analysis revealed higher scores for flavor and taste than for the control sausages; however, it gave them softer texture, which consumers did not like. Similar observations were found by Josquin et al. [37], who reported statistical differences in firmness, fishy taste, and oiliness among Dutch-style fermented sausages manufactured by partial replacement of pork back fat with pure, pre-emulsified, or encapsulated fish oil.

The acceptance rate was also calculated for each formulation of Vienna sausages. As can be seen in Table 4, VSm was the most accepted formulation by consumers (87%), closely followed by VSc (82%), as in the study of Solomando et al. [38] on dry-cured sausages enriched with microencapsulated fish oil, who did not find any significant differences in acceptability between control and enriched batches. VSe, with an acceptance rate of only 53%, failed this test.

Results of the purchase intention testing (see Table 5) show that, for VSe, no panelist ticked the “definitely will buy” and “probably will buy” boxes from the questionnaire; 17% of them answered “definitely will buy” for VSm compared to 13% for VSc. As for the probability of buying, the intention expressed by panelists was higher for VSm (48%) than for VSc (35%); the percentage of undecided ones was 35% for VSc, 26% for VSm, and 9% for VSe. VSe cannot be marketed since 57% of those interviewed responded “definitely will not buy” and 35% “probably will not buy”. The same percentage (4%) of “definitely will not buy” responses were received for VSm and VSe, while of those with “probably will not buy”, 13% was for VSc and 4% for VSe.

All the data discussed above (consumers’ preference, acceptance, and purchase intention) indicate that VSe was disliked by consumers, while the other two formulations were liked equally.

### 3.4. TPC, Polyphenolic Compounds Content, PV, and TBARS Content in Vienna Sausages

The results of TPC determination in Vienna sausage formulations are given in Table 6. The highest TPC was found in VSe (0.27 mg GAE/g), followed by VSm (0.17 mg GAE/g) and VSc (0.10 mg GAE/g), with a statistically significant difference only between VSc and VSe, which means that the use of microcapsules in a concentration of 1.5% in the finished product was not enough to determine a significant polyphenols enrichment effect; however, by replacing ice flakes, an ingredient of 22.9% in the manufacturing recipe of Vienna sausage, with ice cubes from aqueous extract of *Boletus edulis*, the enrichment effect was achieved. The spectrophotometric method for determining TPC is fast, easy, and cheap; however, it cannot identify polyphenolic compounds individually [39]. Therefore, an HPLC method (described in Section 2.4) was further used to estimate them more precisely in Vienna sausages.

The results of chromatographic analysis of polyphenolic compounds in Vienna sausage formulations (see Table 7) revealed instead a significantly higher content of 2,4-dihydroxybenzoic acid, protocatechuic acid, and 1-*O*-galloyl-*β*-D-glucose in VSm and VSe than in VSc; therefore, a polyphenols enrichment effect of Vienna sausages also by using microcapsules. Wang et al. [36] instead have managed to enhance the total phenolic content in sausages by 7.11–34.68 times by substituting the pork lean meat with shiitake mushrooms as such in proportions of 25–100%.

Although the total polyphenol content in the formulation with microcapsules was between those of the other two, the PV and TBARS levels were higher in VSm. This is probably due to the formation of primary and secondary oxidation products in microcapsules during the mushroom extract’s spray drying. The high temperature and rapid dehydration during this treatment oxidize, to some extent, the microencapsulation material [40]. When the TBARS level increases in a food product during storage, it is due to the accumulation of secondary oxidation compounds (aldehydes, ketones, and alcohols) derived from the decomposition of unstable peroxides, the primary oxidation compounds [41]. They may further react and break down, causing a decrease in TBARS level with the following storage times [42].

The TPC did not significantly vary with storage time in any Vienna sausages formulation, framing between 0.10 and 0.13 mg GAE/g in VSc, 0.12 and 0.17 mg GAE/g in VSm, and 0.26 and 0.27 mg GAE/g in VSe (see Table 6). The study by Çam et al. [43] has also demonstrated that microencapsulation provides stability to phenolic compounds during storage; they investigated microencapsulation conditions’ effects on pomegranate peel phenolics’ product quality.

Regarding PV, it was significantly higher in VSm fat at all storage times (1.66–1.84 meq O_2_/kg fat) compared to VSc (1.06–1.18 meq O_2_/kg fat) and VSe (0.95–1.08 meq O_2_/kg fat); it is most likely due to lipid peroxidation in the acidic aqueous extract of *Boletus edulis* mushroom during spray drying to obtain the microcapsules [44]. Although it had the highest lipid content, VSe generally showed the lowest PVs, indicating that the acidic aqueous extract of *Boletus edulis* mushroom protects against lipid peroxidation when used in Vienna sausages.

PV significantly decreased in VSc fat, from an initial level of 1.14 to 1.06 meq O_2_/kg fat on day 3 of storage, then progressively increased until day 8, when it reached a level of 1.18 meq O_2_/kg fat; peroxide levels from the initial day, day 5, and day 8 of storage were not significantly different. TBARS level in VSc showed an upward trend with storage time, significantly increasing from 0.31 mg MDA/kg (initially) to 0.96 mg MDA/kg (day 8 of storage).

In VSm fat, PV did not significantly change until the third storage day, having levels of 1.66 meq O_2_/kg fat (initially) and 1.70 meq O_2_/kg fat (on day 3); from here, it significantly increased to a level of 1.84 meq O_2_/kg fat (day 5), after which it decreased to 1.66 meq O_2_/kg fat (day 8), a non-significant level compared to the initial one and that on third storage day. A significant variation [of increase (day 3: 0.77 mg MDA/kg), decrease (day 5: 0.59 mg MDA/kg), and increase (day 8: 1.23 mg MDA/kg)] was noticed for TBARS level in VSm with storage time, starting from 0.63 mg MDA/kg (initially). In a previous study, Pil-Nam et al. [45] found no significant variation of TBARS during cold storage of frankfurters with added shiitake mushrooms up to 1.2%.

There was a significant increase in PV in VSe fat until the fifth storage day (1.08 meq O_2_/kg fat), from an initial level of 0.95 meq O_2_/kg fat; on the eighth storage day, it reached a level of 1.03 meq O_2_/kg fat, non-significant to that on day 5. As for the TBARS level in VSe, it showed non-significant differences between the initial day (0.63 mg MDA/kg) and day 3 of storage (0.68 mg MDA/kg), with a significant decrease on day 5 (0.54 mg MDA/kg), followed by an increase on day 8 (1.15 mg MDA/kg). No significant correlation was found between PV and TBARS levels in any Vienna sausages formulation.

Given that Vienna sausages were stored for only 8 days in refrigeration conditions, as they are a fresh product, the PV and TBARS levels were much lower (between 0.95 and 1.84 meq O_2_/kg fat for PV, respectively, and between 0.31 and 1.23 mg MDA/kg for TBARS) than the maximum allowed (up to 10 meq O_2_/kg fat for PV in animal fats) [46] or recommended (less than 2 mg MDA/kg for TBARS in pork sausages) [47]. Although the formulation with microcapsules showed higher levels of PV and TBARS than the control one, it received a higher sensory overall score; this is because the consumers can perceive off-flavors and off-odors in meat at MDA levels greater than or equal to 1 mg/kg [48].

## 4. Conclusions

This study’s findings demonstrate that the acidic aqueous extract of *Boletus edulis* mushrooms has a higher polyphenol-enrichment effect on Vienna sausages than the microcapsules obtained from it. However, because of its acidity, the extract affects the emulsion’s stability during the heat treatment of Vienna sausages and, thus, their structure. Moreover, it gives them an unpleasant appearance, a more reddish-yellowish color, a pungent odor, and a sour taste. Consequently, it negatively affects the consumers’ acceptability and purchase intention. All of these inconveniences can be eliminated if the extract is microencapsulated before being used as an ingredient in the preparation of Vienna sausages. A positive aspect of this study is that both the extract and microcapsules do not affect the hardness of Vienna sausage if used in the concentrations tested here.

Thus, it can be concluded that by microencapsulating the acidic aqueous extract of *Boletus edulis* mushrooms and using these microcapsules in the preparation of Vienna sausages, an enrichment effect with polyphenols can be obtained without affecting their technological and sensory properties. Considering the above conclusions, this study can be a starting point for future research on enriching meat products via microencapsulation with polyphenol extracts.

## 5. Patents

Patent Application A/00719 from 8 November 2022: “Process for obtaining and usage of a mushroom extract in Vienna sausages”. Inventors: Melinda Fogarasi, Cristina-Anamaria Semeniuc, Maria-Ioana Socaciu, Dan Cristian Vodnar, and Sonia Ancuța Socaci.

## Figures and Tables

**Table 1 foods-13-00979-t001:** Ingredients used for the preparation of Vienna sausages.

Ingredient	VSc	VSm	VSe
	%		
Cured ground meat	76.4	74.9	76.4
Microcapsules containing acidic aqueous extract of *Boletus edulis* mushrooms	-	1.5	-
Ice flakes	22.9	22.9	-
Ice cubes from acidic aqueous extract of *Boletus edulis* mushrooms	-	-	22.9
Ground black pepper (Trumf International s.r.o., Dolní Újezd, Czech Republic)	0.1	0.1	0.1
Ground nutmeg (Trumf International s.r.o., Dolní Újezd, Czech Republic)	0.1	0.1	0.1
Sweet pepper paprika (Paprika Kalocsa 180; Trumf International s.r.o., Dolní Újezd, Czech Republic)	0.2	0.2	0.2
Garlic granules (S.C. Ion Moș S.R.L., Chiajna, Romania)	0.2	0.2	0.2
Polyphosphates (Brätfix; Frutarom Savory Solutions, Salzburg, Austria)	0.2	0.2	0.2
Total	100.0	100.0	100.0

VSc, control Vienna sausages; VSm, Vienna sausages enriched with 1.5% microcapsules containing acidic aqueous extract of *Boletus edulis* mushrooms; VSe, Vienna sausages enriched with acidic aqueous extract of *Boletus edulis* mushrooms.

**Table 2 foods-13-00979-t002:** Proximate composition, energy value, pH, and EHN content in Vienna sausages at various storage times (T1, T2, T3, and T4).

Parameter/ Energy Value	VSc				VSm				VSe			
	T1	T2	T3	T4	T1	T2	T3	T4	T1	T2	T3	T4
Moisture (%)	64.40 ± 0.104 ^aA^	63.12 ± 0.073 ^bA^	63.85 ± 0.117 ^aA^	62.14 ± 0.221 ^cA^	62.97 ± 0.275 ^aB^	62.01 ± 0.446 ^aB^	62.70 ± 0.247 ^aB^	61.71 ± 0.572 ^aA^	59.84 ± 0.025 ^bC^	58.77 ± 0.026 ^aC^	60.55 ± 0.281 ^cC^	58.90 ± 0.119 ^cB^
Protein (%)	15.02 ± 0.169 ^bC^	16.30 ± 0.045 ^aB^	15.55 ± 0.212 ^bB^	16.47 ± 0.005 ^aB^	16.28 ± 0.058 ^aB^	16.19 ± 0.044 ^aB^	15.93 ± 0.105 ^aB^	16.12 ± 0.236 ^aB^	18.84 ± 0.005 ^aA^	18.80 ± 0.026 ^aA^	18.45 ± 0.574 ^aA^	19.26 ± 0.003 ^aA^
Fat (%)	9.84 ± 0.375 ^aB^	7.23 ± 0.173 ^bB^	9.45 ± 0.404 ^aB^	8.94 ± 0.283 ^aC^	7.08 ± 0.244 ^cC^	6.79 ± 0.125 ^cB^	8.75 ± 0.385 ^bB^	10.60 ± 0.051 ^aB^	15.28 ± 0.302 ^aA^	14.73 ± 0.139 ^aA^	15.50 ± 0.090 ^aA^	15.34 ± 0.298 ^aA^
Ash (%)	1.44 ± 0.040 ^aA^	1.53 ± 0.099 ^aA^	1.37 ± 0.163 ^aA^	1.41 ± 0.089 ^aA^	1.50 ± 0.141 ^aA^	1.55 ± 0.148 ^aA^	1.30 ± 0.100 ^aA^	1.51 ± 0.245 ^aA^	1.61 ± 0.146 ^abA^	1.80 ± 0.040 ^aA^	1.36 ± 0.049 ^bA^	1.63 ± 0.029 ^abA^
Total carbohydrate (%)	9.29 ± 0.141 ^cB^	11.82 ± 0.302 ^aB^	9.77 ± 0.570 ^bcA^	11.03 ± 0.155 ^abA^	12.11 ± 0.312 ^abA^	13.46 ± 0.424 ^aA^	11.32 ± 0.638 ^bA^	10.06 ± 0.632 ^bA^	4.41 ± 0.458 ^bC^	5.90 ± 0.099 ^aC^	4.15 ± 0.252 ^bB^	4.87 ± 0.385 ^abB^
EV (kcal/100 g)	186 ± 2.132 ^aB^	178 ± 0.176 ^bB^	186 ± 2.199 ^aB^	190 ± 1.943 ^aC^	177 ± 3.214 ^bB^	180 ± 3.003 ^bB^	188 ± 1.333 ^bB^	200 ± 3.012 ^aB^	231 ± 0.868 ^bA^	231 ± 0.962 ^abA^	230 ± 0.476 ^bA^	235 ± 1.128 ^aA^
pH	6.40 ± 0.018 ^aA^	6.39 ± 0.003 ^aA^	6.37 ± 0.016 ^aA^	6.41 ± 0.005 ^aA^	6.26 ± 0.011 ^bB^	6.36 ± 0.012 ^aA^	6.34 ± 0.006 ^aA^	6.39 ± 0.016 ^aA^	4.94 ± 0.006 ^aC^	5.00 ± 0.019 ^aB^	4.97 ± 0.028 ^aB^	4.96 ± 0.007 ^aB^
EHN (mg NH_3_/100 g)	13.60 ± 0.001 ^cB^	16.03 ± 1.115 ^bB^	20.19 ± 0.033 ^aB^	15.27 ± 0.003 ^bcB^	18.97 ± 0.008 ^bA^	22.73 ± 0.937 ^aA^	23.75 ± 0.003 ^aA^	22.10 ± 0.001 ^aA^	18.53 ± 0.201 ^bA^	21.10 ± 1.017 ^aA^	18.68 ± 0.001 ^bC^	22.27 ± 0.261 ^aA^

VSc, control Vienna sausages; VSm, Vienna sausages enriched with 1.5% microcapsules containing acidic aqueous extract of *Boletus edulis* mushrooms; VSe, Vienna sausages enriched with acidic aqueous extract of *Boletus edulis* mushrooms; T1, initial; T2, day 3 of storage; T3, day 5 of storage; T4, day 8 of storage; EV, energy value; EHN, easily hydrolyzable nitrogen. Data are expressed as mean ± standard deviation values of six measurements. Different lowercase letters within a row indicate significant differences between storage times (*p* < 0.05, Tukey’s test), and different uppercase letters show significant differences between Vienna sausage formulations (*p* < 0.05).

**Table 3 foods-13-00979-t003:** Color attributes of Vienna sausages at various storage times (T1, T2, T3, and T4).

Color Attribute	VSc				VSm				VSe			
	Initial	Day 3	Day 5	Day 8	Initial	Day 3	Day 5	Day 8	Initial	Day 3	Day 5	Day 8
*L**	70.86 ± 0.759 ^aB^	69.83 ± 1.094 ^abB^	70.12 ± 1.392 ^aA^	68.82 ± 0.809 ^bB^	72.15 ± 0.046 ^aA^	72.43 ± 0.914 ^aA^	72.08 ± 0.778 ^aA^	71.80 ± 0.602 ^aA^	64.73 ± 1.187 ^abC^	65.34 ± 1.137 ^aC^	62.55 ± 3.272 ^bcB^	61.57 ± 2.394 ^cC^
*a**	10.29 ± 0.446 ^bB^	10.78 ± 0.561 ^aC^	10.30 ± 0.141 ^bB^	10.25 ± 0.276 ^bB^	9.68 ± 0.259 ^cC^	9.97 ± 0.153 ^aA^	9.89 ± 0.228 ^bcC^	9.95 ± 0.228 ^bC^	16.69 ± 0.490 ^aA^	16.48 ± 0.456 ^abB^	15.99 ± 0.506 ^bA^	16.04 ± 0.368 ^bA^
*b**	17.68 ± 0.532 ^aC^	17.44 ± 0.373 ^aC^	17.35 ± 0.237 ^aB^	17.71 ± 0.519 ^aC^	18.93 ± 0.336 ^cB^	19.51 ± 0.203 ^abB^	19.67 ± 0.524 ^aA^	19.21 ± 0.411 ^bcB^	19.98 ± 0.926 ^aA^	20.75 ± 0.803 ^aC^	19.70 ± 1.346 ^aA^	21.48 ± 0.919 ^aA^
*h**	1.04 ± 0.012 ^aB^	1.02 ± 0.018 ^bB^	1.03 ± 0.005 ^aB^	1.05 ± 0.008 ^aB^	1.10 ± 0.014 ^aA^	1.10 ± 0.006 ^aA^	1.10 ± 0.012 ^aA^	1.09 ± 0.013 ^aA^	0.87 ± 0.028 ^bC^	0.90 ± 0.026 ^abC^	0.89 ± 0.036 ^bC^	0.93 ± 0.024 ^aC^
*C**	20.45 ± 0.635 ^aC^	20.51 ± 0.562 ^aC^	20.18 ± 0.255 ^aC^	20.46 ± 0.562 ^aC^	21.27 ± 0.310 ^bB^	21.91 ± 0.212 ^aB^	22.02 ± 0.510 ^aB^	21.64 ± 0.372 ^abB^	26.05 ± 0.759 ^abA^	26.50 ± 0.607 ^aA^	25.39 ± 1.094 ^bA^	26.81 ± 0.745 ^aA^
Δ*E*	-	-	-	-	2.07	3.55	3.32	3.51	9.29	8.11	10.02	10.23

VSc, control Vienna sausages; VSm, Vienna sausages enriched with 1.5% microcapsules containing acidic aqueous extract of *Boletus edulis* mushrooms; VSe, Vienna sausages enriched with acidic aqueous extract of *Boletus edulis* mushrooms; *L**, lightness; *a**, redness; *b**, yellowness; *h**, hue angle; *C**, chroma, Δ*E*, total color difference. Δ*E* between 0 and 0.5 = a color difference at the trace level; Δ*E* between 0.5 and 1.5 = a slight color difference; Δ*E* between 1.5 and 3.0 = a noticeable color difference; Δ*E* between 3.0 and 6.0 = an appreciable color difference; Δ*E* between 6 and 12.0 = a large color difference; Δ*E* higher than 12.0 = an obvious color difference. Data are expressed as mean ± standard deviation values of twenty-four measurements. Different lowercase letters within a row indicate significant differences between storage times (*p* < 0.05), and different uppercase letters show significant differences between Vienna sausage formulations (*p* < 0.05).

**Table 4 foods-13-00979-t004:** Hedonic scores for Vienna sausage’s sensory attributes and acceptance rates.

Formulation	Sensory Attributes							Acceptance Rate (%)
	Appearance	Color	Odor	Taste	Texture	Overall Acceptability	Overall Score	
VSc	7.3 ± 0.974 ^A^	6.8 ± 1.380 ^A^	7.6 ± 1.270 ^A^	7.6 ± 1.438 ^A^	7.7 ± 1.191 ^A^	7.6 ± 1.121 ^A^	7.4 ± 0.823 ^A^	82
VSm	7.5 ± 1.082 ^A^	7.4 ± 1.308 ^A^	7.6 ± 1.270 ^A^	7.9 ± 1.125 ^A^	8.0 ± 0.878 ^A^	8.0 ± 1.107 ^A^	7.8 ± 0.861 ^A^	87
VSe	3.8 ± 1.899 ^B^	4.4 ± 1.828 ^B^	5.8 ± 2.387 ^B^	4.7 ± 2.397 ^B^	4.9 ± 2.096 ^B^	5.0 ± 2.011 ^B^	4.8 ± 1.850 ^B^	53

VSc, control Vienna sausages; VSm, Vienna sausages enriched with 1.5% microcapsules containing acidic aqueous extract of *Boletus edulis* mushrooms; VSe, Vienna sausages enriched with acidic aqueous extract of *Boletus edulis* mushrooms. Data are expressed as mean ± standard deviation values of eighty-two responses. Different uppercase letters within a column indicate significant differences between Vienna sausage formulations (*p* < 0.05, Tukey’s test).

**Table 5 foods-13-00979-t005:** Response rates (%) for purchase intention of Vienna sausages.

Formulation	Definitely Will Buy	Probably Will Buy	Might or Might Not Buy	Probably Will Not Buy	Definitely Will Not Buy
VSc	13.04	34.78	34.78	13.04	4.35
VSm	17.39	47.82	26.09	4.35	4.35
VSe	-	-	8.70	34.78	56.52

VSc, control Vienna sausages; VSm, Vienna sausages enriched with 1.5% microcapsules containing acidic aqueous extract of *Boletus edulis* mushrooms; VSe, Vienna sausages enriched with acidic aqueous extract of *Boletus edulis* mushrooms.

**Table 6 foods-13-00979-t006:** TPC, PV, and TBARS content in Vienna sausages at various storage times (T1, T2, T3, and T4).

Color Attribute	VSc				VSm				VSe			
	Initial	Day 3	Day 5	Day 8	Initial	Day 3	Day 5	Day 8	Initial	Day 3	Day 5	Day 8
TPC (mg GAE/g)	0.10 ± 0.034 ^aB^	0.11 ± 0.010 ^aA^	0.11 ± 0.006 ^aC^	0.13 ± 0.008 ^aB^	0.17 ± 0.037 ^aAB^	0.13 ± 0.003 ^aA^	0.16 ± 0.0 ^aB^	0.12 ± 0.024 ^aB^	0.27 ± 0.037 ^aA^	0.26 ± 0.069 ^aA^	0.27 ± 0.007 ^aA^	0.26 ± 0.030 ^aA^
PV (meq O_2_/kg fat)	1.14 ± 0.004 ^aB^	1.06 ± 0.006 ^bB^	1.16 ± 0.008 ^aB^	1.18 ± 0.018 ^aB^	1.66 ± 0.030 ^bA^	1.70 ± 0.029 ^bA^	1.84 ± 0.020 ^aA^	1.66 ± 0.016 ^bA^	0.95 ± 0.024 ^cC^	1.00 ± 0.022 ^bcB^	1.08 ± 0.017 ^aC^	1.03 ± 0.001 ^abC^
TBARS (mg MDA/kg)	0.31 ± 0.017 ^dB^	0.39 ± 0.007 ^cB^	0.57 ± 0.003 ^bA^	0.96 ± 0.003 ^aC^	0.63 ± 0.007 ^cA^	0.77 ± 0.034 ^bA^	0.59 ± 0.017 ^cA^	1.23 ± 0.014 ^aA^	0.63 ± 0.010 ^bA^	0.68 ± 0.017 ^bA^	0.54 ± 0.017 ^cA^	1.15 ± 0.010 ^aB^

VSc, control Vienna sausages; VSm, Vienna sausages enriched with 1.5% microcapsules containing acidic aqueous extract of *Boletus edulis* mushrooms; VSe, Vienna sausages enriched with acidic aqueous extract of *Boletus edulis* mushrooms; TPC, total phenolic content; PV, peroxide value; TBARS, thiobarbituric acid reactive substances. Data are expressed as mean ± standard deviation values of six measurements. Different lowercase letters within a row indicate significant differences between storage times (*p* < 0.05), and different uppercase letters show significant differences between Vienna sausage formulations (*p* < 0.05).

**Table 7 foods-13-00979-t007:** Content of polyphenolic compounds (μg/g) in Vienna sausages.

Crt. No.	Compound	Chemical Class	Chemical Subclass	VSc	VSm	VSe
1	2-Hydroxybenzoic acid	PAs	HBA_S_	245.43 ± 22.240 ^A^	283.34 ± 6.824 ^A^	313.28 ± 34.453 ^A^
2	2,4-Dihydroxybenzoic acid	PAs	HBA_S_	71.75 ± 2.923 ^B^	84.47 ± 3.490 ^A^	85.32 ± 0.040 ^A^
3	Gallic acid	PAs	HBA_S_	113.93 ± 3.809 ^A^	119.82 ± 5.232 ^A^	135.62 ± 24.465 ^A^
4	Syringic acid	PAs	HBA_S_	5.00 ± 0.248 ^A^	5.29 ± 0.523 ^A^	5.80 ± 3.828 ^A^
5	Protocatechuic acid	PAs	HBA_S_	8.97 ± 0.528 ^B^	17.07 ± 0.555 ^A^	15.41 ± 0.673 ^A^
6	1-*O*-Galloyl-*β*-D-glucose	PAs	HBA_S_	0.69 ± 0.081 ^B^	2.61 ± 0.110 ^A^	2.47 ± 0.514 ^A^
7	4-Hydroxybenzoic acid	PAs	HBA_S_	2.21 ± 1.515 ^A^	3.24 ± 0.236 ^A^	3.21 ± 0.525 ^A^
8	Quercetin 3-*O*-acetyl-rhamnoside	FVs	FVols	1.73 ± 0.093 ^A^	1.72 ± 0.127 ^A^	1.74 ± 0.154 ^A^
9	Quercetin 3-*O*-rutinoside (Rutin)	FVs	FVols	2.00 ± 0.217 ^A^	1.92 ± 0.004 ^A^	2.11 ± 0.264 ^A^
10	Quercetin 3-*O*-malonyl-glucoside	FVs	FVols	1.86 ± 0.182 ^A^	1.83 ± 0.025 ^A^	2.14 ± 0.472 ^A^
11	Quercetin 3-*O*-glucosyl-rhamnosyl-glucoside	FVs	FVols	1.85 ± 0.373 ^A^	1.70 ± 0.049 ^A^	1.83 ± 0.131 ^A^
	Total content			455.41	523.03	568.92

VSc, control Vienna sausages; VSm, Vienna sausages enriched with 1.5% microcapsules containing acidic aqueous extract of *Boletus edulis* mushrooms; VSe, Vienna sausages enriched with acidic aqueous extract of *Boletus edulis* mushrooms; PAs, phenolic acids; FVs, flavonoids; HBAs, hydroxybenzoic acids; FVols, flavonols. Data are expressed as mean ± standard deviation values of six measurements. Different uppercase letters within a row indicate significant differences between Vienna sausage formulations (*p* < 0.05, Tukey’s test).

## Data Availability

The original contributions presented in the study are included in the article, further inquiries can be directed to the corresponding author.

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
