# Peer review of "Polyphenols-Enrichment of Vienna Sausages Using Microcapsules Containing Acidic Aqueous Extract of Boletus edulis Mushrooms"

_foods, 2024, doi:10.3390/foods13070979_

Round 1

Reviewer 1 Report

Comments and Suggestions for Authors

Major comment:

1. There are doubts about the caloric analysis in 2.4. Since microcrystalline cellulose cannot be metabolized in the human body to produce energy, including it in the calories will overestimate the calories. It is recommended to deduct its content before calculating.

2. Vienna Sausages are emulsified sausages, which need to be finely chopped and emulsified during the production process to achieve homogeneous properties. In 2.3. it is also mentioned that the 9 kg of pork is finely chopped. However, it was mentioned in the results and discussion that data differences may be caused by different proportions of fat in pork tissue. This explanation is unreasonable, and you should confirm whether the description in the manuscript is correct.

3. The 9 kg of pork is mixed with only 0.18 g of salt, which does not meet the meat processing ratio. Please confirm whether the added amount is correct.

4. It is recommended that T1-T4 of Table 2, 3, and 6. be changed to day 0, 3, 5 and 8. Tukey’s test has been described in 2.5., and there is no need to write comments in the table.

5. What ingredient does Line 232: Sour taste come from? should be discussed.

6. Volatile basic nitrogen (VBN) may be produced during the storage process of the sample, but this article only discusses it in terms of ammonia and is incomplete.

7. Sensory analysis does not discuss whether the score range is a 9-point method, which should be supplemented in the manuscript.

8. The acceptance rate below which is failed should be stated in the research method and should be well-founded.

9. According to the data in Table 6, the polyphenols in Mushroom did not produce antioxidant benefits and even increased the degree of lipid peroxidation. The actual proportion of acetic acid in the VSe liquid is too high, and its content can significantly change the flavor of the product. The necessity of this set of experiments cannot be demonstrated.

10. There is no need to add ‘*’ to ΔE, and the ΔE value at which significant differences can be observed by humans should be listed.

Author Response

COMMENTS FOR THE AUTHOR:

Reviewer #1: 10 comments

Response to the Reviewer 1

Dear Reviewer,

We thank you for your comments and suggestions, which allowed us to improve the manuscript's quality considerably. We agree with all your comments and corrected the manuscript point by point accordingly. Our changes are marked in red in the revised manuscript using the "Track Changes" feature of Microsoft Word. Please find our answers below.

Comment 1. There are doubts about the caloric analysis in 2.4. Since microcrystalline cellulose cannot be metabolized in the human body to produce energy, including it in the calories will overestimate the calories. It is recommended to deduct its content before calculating.

Answer: We did not determine the cellulose in Vienna sausages (those with microcapsule and the acid extract) to be able to extract this parameter from the total carbohydrate content. We mentioned the calculation formulas used to estimate the energy value of Vienna sausages in lines 187-189 of the manuscript. Other authors also used these in studies of this kind, and we cited them in this manuscript.

Comment 2. Vienna Sausages are emulsified sausages, which need to be finely chopped and emulsified during the production process to achieve homogeneous properties. In 2.3. it is also mentioned that the 9 kg of pork is finely chopped. However, it was mentioned in the results and discussion that data differences may be caused by different proportions of fat in pork tissue. This explanation is unreasonable, and you should confirm whether the description in the manuscript is correct.

Answer: The pork chunks are finely chopped through a 3-mm sieve, enough for the homogenous distribution of fat deposited between muscle bundles but not enough for the homogenous distribution of fat deposited between the muscle fibres (of the order of micrometres). Moreover, this mass of meat was distributed in three portions before being finely chopped and emulsified (with the rest of the ingredients) to prepare the Vsc, Vsm, and Vse; hence, the compositional differences between the Vienna sausage formulations. Please see lines 158-162 in the revised manuscript.

We thank you for this observation, which another reviewer also made. Therefore, we added additional explanations in the manuscript: "Pork meat contains muscle, connective, adipose, vascular, and nervous tissues [Listrat et al., 2016]; their proportions vary with the slaughtered individual's fattening state [Arshad et al., 2018]. Some meat cuts may have more fat, and the fat deposition between muscle fibers and bundles (fascicles) is not uniform [Schumacher et al., 2022]. The pork chunks used as raw material to prepare the Vienna sausage may originate from more porcine animals, hence the compositional differences between the Vienna sausage formulations." Please see lines 298-303 in the revised manuscript.

Comment 3. The 9 kg of pork is mixed with only 0.18 g of salt, which does not meet the meat processing ratio. Please confirm whether the added amount is correct.

Answer: We thank you for this observation. It is 0.18 kg, which we corrected. Please see line 157 in the revised manuscript.

Comment 4. It is recommended that T1-T4 of Table 2, 3, and 6. be changed to day 0, 3, 5 and 8. Tukey’s test has been described in 2.5., and there is no need to write comments in the table.

Answer: We thank you for this observation. We made the changes you suggested. Please see the revised form of the manuscript.

Comment 5. What ingredient does Line 232: Sour taste come from? should be discussed.

Answer: The acidic aqueous extract of Boletus edulis mushrooms (used as such instead of ice flakes) to formulate the VSe.

Comment 6. Volatile basic nitrogen (VBN) may be produced during the storage process of the sample, but this article only discusses it in terms of ammonia and is incomplete.

Answer: We thank you for this observation, which another reviewer also made. This test quantifies, in mg NH3/100 g, the ammonia, amines, and other alkaline substances formed in Vienna sausages due to the protein decomposition caused by microbial spoilage. We added this information to the manuscript. Please see lines 364-366.

Comment 7. Sensory analysis does not discuss whether the score range is a 9-point method, which should be supplemented in the manuscript.

Answer: We thank you for this observation. We added the following information to the manuscript, "The 9-point hedonic scale test was used to evaluate the appearance, color, odor, taste, texture, and overall acceptability of Vienna sausages." Please see lines 208-209 in the revised manuscript.

Comment 8. The acceptance rate below which is failed should be stated in the research method and should be well-founded.

Answer: We thank you for this observation. We added the following information in the manuscript, "The acceptance rate of each Vienna sausage formulation was calculated using the following Formula (1):

where: X is the formulation's mean sensory score, and N is the maximum sensory score it received. An acceptance rate equal to or greater than 70% is considered reasonable [19]". Please see lines 214-219 in the revised manuscript.

Comment 9. According to the data in Table 6, the polyphenols in Mushroom did not produce antioxidant benefits and even increased the degree of lipid peroxidation. The actual proportion of acetic acid in the VSe liquid is too high, and its content can significantly change the flavor of the product. The necessity of this set of experiments cannot be demonstrated.

Answer: We thank you for this observation, which another reviewer also made. We added additional explanations related to these aspects to the manuscript.

The spectrophotometric method for determining TPC is fast, easy, and cheap; however, it cannot identify polyphenolic compounds individually [López-Fernández et al., 2020]. Therefore, the HPLC method described in subsection 2.4 was further used to estimate them more precisely in Vienna sausages. Please see lines 551-554. That revealed a significantly higher content of some constituents in the enriched formulations than in the control one, thus a higher total content (please see Table 7). However, the PV and TBARS levels were higher in the formulation with microcapsules. This is probably due to the formation of primary and secondary oxidation products in microcapsules during the mushroom extract's spray drying. The high temperature and rapid dehydration during this treatment oxidize, to some extent, the microencapsulation material [Xiong et al., 2021]. Please see lines 593-602.

The use of extract as such gave the Vienna sausages an unpleasant appearance, pungent odor, and sour taste (please see lines 446). For this reason, it received the lowest score (4.8 points for overall score) in the sensory analysis and an acceptance rate of only 53%. In conclusion, microcapsules are more suitable for use as a polyphenol enrichment ingredient in Vienna sausages than the extract. Please see lines 33-35 in the revised manuscript.

Comment 10. There is no need to add ‘*’ to ΔE, and the ΔE value at which significant differences can be observed by humans should be listed.

Answer: We thank you for this observation. We deleted the asterisk from ΔE. We added the following information below Table 3, "ΔE between 0 and 0.5 = a color difference at the trace level; ΔE between 0.5 and 1.5 = a slight color difference; ΔE between 1.5 and 3.0 = a noticeable color difference; ΔE between 3.0 and 6.0 = an appreciable color difference; ΔE between 6−12.0 = a large color difference; ΔE higher than 12.0 = an obvious color difference." Please see the revised form of the manuscript.

Reviewer 2 Report

Comments and Suggestions for Authors

This study evaluated the storage stability of Vienna sausages with and without Boletus edulis mushroom extract (encapsulated or not). The perspective presented in this study is unique and deserves further investigation. It was a well-written article. However, there are still some questions needed to be considered.

1. Abstract need to be revised. There are many results of the manuscript that have not been summarized. In addition, the storage stability of Vienna sausages is also a focus of the research.

2. Line 175. p<0.05

3. Why prepare microcapsules instead of others?

4. Can the authors explain why is the ratio of microcapsules 1.5%? Will the properties of Vienna sausages improve with a higher ratio?

5. Section 3.1. The description of the results for moisture, protein, fat, ash, pH, etc. could be written according to factors or according to comparing Vienna sausage formulations first and then comparing storage time. What is the significance of the change of the content of every factor?

6. Line 212-215. Can the non-homogeneous distribution of fat between the muscle fibers (marbling) of pork meat lead to a significant difference in fat content?

7. Line 263-264. What I see is different. The difference between the initial energy value and the final one of VSm was higher than at the other two formulations.

8. Whats the relationship of physicochemical properties of Vienna Sausages and sensory or texture?

9. Line 400-407. Why the polyphenol enrichment effect of VSe was more significant than that of VSm?

10. I think it more intuitive to translate tables 2, 3, and 6 into figures.

11. How to solve the problem of lipid peroxidation of VSm?

12. What do the increase and decrease of TBARS represent? Please explain the results rather than just describing the trends.

13. It can be seen from the abstract, conclusion and introduction that enriching polyphenols is the main purpose of the study, but the result of TPC revealed that polyphenol enrichment effect of VSe was more significant than that of VSm. I think it is better to emphasize in the abstract and conclusion the difference between VSm and VSe.

Author Response

COMMENTS FOR THE AUTHOR:

Reviewer #1: This study evaluated the storage stability of Vienna sausages with and without Boletus edulis mushroom extract (encapsulated or not). The perspective presented in this study is unique and deserves further investigation. It was a well-written article. However, there are still some questions needed to be considered.

Response to the Reviewer 1

Dear Reviewer,

We thank you for your comments and suggestions, which allowed us to improve the manuscript's quality considerably. We agree with all your comments and corrected the manuscript point by point accordingly. Our changes are marked in red in the revised manuscript using the "Track Changes" feature of Microsoft Word. Please find our answers below.

Comment 1. Abstract need to be revised. There are many results of the manuscript that have not been summarized. In addition, the storage stability of Vienna sausages is also a focus of the research.

Answer: We thank you for this observation. The abstract was revised as suggested. Please see lines 17-35 in the revised manuscript.

Comment 2. Line 175. p<0.05

Answer: We thank you for this observation. We italicized the p. Please see line 236 in the revised manuscript.

Comment 3. Why prepare microcapsules instead of others?

Answer: Polyphenols can be successfully extracted from Boletus edulis mushrooms with acidic water. However, the extract may influence the organoleptic or textural properties of the product to which it is added for enrichment; this inconvenience can be avoided by microencapsulating it using spray drying. Please see lines 17-21 in the revised manuscript.

Comment 4. Can the authors explain why is the ratio of microcapsules 1.5%? Will the properties of Vienna sausages improve with a higher ratio?

Answer: The volume of the microcapsules obtained by spray drying is large. The more these are added to the product, the more intense the polyphenol enrichment effect will be, but it will affect its texture. At the same time, the production cost of Vienna sausages will also increase.

Figure 1. Microcapsules                                            Figure 2. Matured meat with the added microcapsules

Comment 5. Section 3.1. The description of the results for moisture, protein, fat, ash, pH, etc. could be written according to factors or according to comparing Vienna sausage formulations first and then comparing storage time. What is the significance of the change of the content of every factor?

Answer: In section 3.1, we discussed in detail, for each parameter separately, first the effect of formulation (please see lines 242-312) and then the impact of the storage time (please see lines 313-352).

Comment 6. Line 212-215. Can the non-homogeneous distribution of fat between the muscle fibers (marbling) of pork meat lead to a significant difference in fat content?

Answer: Pork meat contains muscle, connective, adipose, vascular, and nervous tissues [Listrat et al., 2016]; their proportions vary with the slaughtered individual's fattening state [Arshad et al., 2018]. Some meat cuts may have more fat than others, and the fat deposition between muscle bundles is not uniform [Schumacher et al., 2022]. The pork chunks used as raw material to prepare the Vienna sausage may originate from more porcine animals, hence the compositional differences between the Vienna sausage formulations. Please see lines 290-295 in the revised manuscript.

Comment 7. Line 263-264. What I see is different. The difference between the initial energy value and the final one of VSm was higher than at the other two formulations.

Answer: The slightly higher energy values at the end of the storage period are mainly due to the water losses in Vienna sausage formulations (please see lines 340-350). The VSm sample did not lose as much water during storage as the other two formulations.

Comment 8. What’s the relationship of physicochemical properties of Vienna Sausages and sensory or texture?

Answer: The use of extract as such gave the Vienna sausages an unpleasant appearance, pungent odor, and sour taste (please see lines 449-451). For this reason, it received the lowest score (4.8 points for overall score) in the sensory analysis and an acceptance rate of only 53%.

Comment 9. Line 400-407. Why the polyphenol enrichment effect of VSe was more significant than that of VSm?

Answer: This is because mushroom extract, which represents 22.9% of the manufacturing recipe, was used instead of water to prepare the sausages. In VSm, 1.5% of the microcapsules were used.

Comment 10. I think it more intuitive to translate tables 2, 3, and 6 into figures.

Answer: We thank you very much for this suggestion. However, this would mean making a graph for each parameter, which, considering the number of determined parameters, would mean a lot of graphs. Therefore, in this case, we believe that showing the results in tables is more appropriate.

Comment 11. How to solve the problem of lipid peroxidation of VSm?

Answer: Given that Vienna sausages were stored for only 8 days in refrigeration conditions, as they are a fresh product, the PV and TBARS levels were much lower (between 0.95 and 1.84 meq O2/kg fat for PV, respectively, between 0.31 and 1.23 mg MDA/kg for TBARS) than the maximum allowed (up to 10 meq O2/kg fat for PV in animal fats [CXS 211-1999] or recommended (less than 2 mg MDA/kg for TBARS in pork sausages) [Wenjiao et al., 2014]. Although the formulation with microcapsules showed higher levels of PV and TBARS than the control one, it received a higher sensory overall score; that is because the consumers can perceive off-flavours and off-odours in meat at MDA levels greater than or equal to 1 mg/kg [Draszanowska et al., 2022]. Please see lines 575-582 in the revised manuscript.

Comment 12. What do the increase and decrease of TBARS represent? Please explain the results rather than just describing the trends.

Answer: The increase in TBARS level in Vienna sausages during storage is due to the accumulation of secondary oxidation compounds (aldehydes, ketones, and alcohols) derived from the decomposition of unstable peroxides, the primary oxidation compounds [Chu et al., 2023]. They may further react and break down, causing a decrease in TBARS level with the following storage times [Oduor-Odote and Obiero, 2009]. Please see lines 529-539.

Comment 13. It can be seen from the abstract, conclusion and introduction that enriching polyphenols is the main purpose of the study, but the result of TPC revealed that polyphenol enrichment effect of VSe was more significant than that of VSm. I think it is better to emphasize in the abstract and conclusion the difference between VSm and VSe.

Answer: We thank you for this observation. In the abstract and conclusion, we emphasized the difference between VSm and VSe. Please see the revised manuscript.

Another 13 new references were introduced in the revised manuscript.

Reviewer 3 Report

Comments and Suggestions for Authors

The paper studies the effect of acidic extract from Boletus edulis mushroom incorporated directly or encapsulated in Vienna Sausages on physicochemical, sensorial, and oxidation parameters during cold storage. The paper presents relevant information and the introduction, material, and methods sections are well-written allowing the reproducibility of the test. However, the objective should clearly be described at the end of the introduction section.  The main weaknesses of the paper are in the discussion of the results. The authors describe the obtained results but I miss a better interpretation of them. How encapsulated extract provide good textural and sensorial numbers but present a higher oxidation than even the control sample? Also, the sample with the acidic extract has good oxidation values but it is not sensorially preferred. How can explain the authors the differences in composition between samples? So, I would recommend improving the discussion of the results and establishing hypothesis with references about what happened in the samples. 

I have included a list of several comments:

Line 55-67 seem like an abstract with information that should be included in the material and methods. The objective of the work should be clearly described at the end of the introduction section

Line 127: it should be indicated that the storage time variable should be analyzed also and the analysis days should be included. 

Line 147: Information about illuminant and observer angle should be included.

Line 172: PV? Means?

Line 192: The diminution of moisture and the increase in protein content in VSe is quite surprising and it should be better explained. I do not believe is only because of the high substitution… The composition of acidic aqueous extract was measured¿? Line 220. Why? What kind of compounds were extracted with the solution apart from polyphenols?

Line 193: It is not in contrast since you have a decrease of moisture as well as Kanwal et al. 

Line 200-206: Check the style as a footnote of the table.

Line 227: Why composition values should change along the storage period and also the energy value? Were the Vienna sausages packaged? This information should be included in the material and methods. 

Line 269: Could there have been microbial spoilage responsible for the pH increase?

Line 408: table 6. Quite surprising that peroxide values and TBARS are higher in samples with encapsulated extract and low polyphenol contents. Could be related to the extraction of polyphenols from the matrix with the encapsulated extracts? A kind of underestimation? I miss a better discussion of the results rather than a description. 

Comments on the Quality of English Language

The English is ok, minor editing. 

Author Response

COMMENTS FOR THE AUTHOR:

Reviewer #2: The paper studies the effect of acidic extract from Boletus edulis mushroom incorporated directly or encapsulated in Vienna Sausages on physicochemical, sensorial, and oxidation parameters during cold storage. The paper presents relevant information and the introduction, material, and methods sections are well-written allowing the reproducibility of the test. However, the objective should clearly be described at the end of the introduction section. The main weaknesses of the paper are in the discussion of the results. The authors describe the obtained results but I miss a better interpretation of them. How encapsulated extract provide good textural and sensorial numbers but present a higher oxidation than even the control sample? Also, the sample with the acidic extract has good oxidation values but it is not sensorially preferred. How can explain the authors the differences in composition between samples? So, I would recommend improving the discussion of the results and establishing hypothesis with references about what happened in the samples.

Response to the Reviewer 2

Dear Reviewer,

We thank you for your comments and suggestions, which allowed us to improve the manuscript's quality considerably. We agree with all your comments and corrected the manuscript point by point accordingly. Our changes are marked in red in the revised manuscript using the "Track Changes" feature of Microsoft Word. Please find our answers below.

I have included a list of several comments:

Comment 1. How encapsulated extract provide good textural and sensorial numbers but present a higher oxidation than even the control sample?

Answer: Given that Vienna sausages were stored for only 8 days in refrigeration conditions, as they are a fresh product, the PV and TBARS levels were much lower (between 0.95 and 1.84 meq O2/kg fat for PV, respectively, between 0.31 and 1.23 mg MDA/kg for TBARS) than the maximum allowed (up to 10 meq O2/kg fat for PV in animal fats [CXS 211-1999] or recommended (less than 2 mg MDA/kg for TBARS in pork sausages) [Wenjiao et al., 2014]. Although the formulation with microcapsules showed higher levels of PV and TBARS than the control one, it received a higher sensory overall score; that is because the consumers can perceive off-flavours and off-odours in meat at MDA levels greater than or equal to 1 mg/kg [Draszanowska et al., 2022]. Please see lines 575-582 in the revised manuscript.

Comment 2. Also, the sample with the acidic extract has good oxidation values but it is not sensorially preferred.

Answer: The use of extract as such gave the Vienna sausages an unpleasant appearance, pungent odor, and sour taste (please see lines 449-451). For this reason, it received the lowest score (4.8 points for overall score) in the sensory analysis and an acceptance rate of only 53%.

Comment 3. How can explain the authors the differences in composition between samples?

Pork meat contains muscle, connective, adipose, vascular, and nervous tissues [Listrat et al., 2016]; their proportions vary with the slaughtered individual's fattening state [Arshad et al., 2018]. Some meat cuts may have more fat than others, and the fat deposition between muscle bundles is not uniform [Schumacher et al., 2022]. The pork chunks used as raw material to prepare the Vienna sausage may originate from more porcine animals, hence the compositional differences between the Vienna sausage formulations. Please see lines 290-295 in the revised manuscript.

Comment 4. Line 55-67 seem like an abstract with information that should be included in the material and methods. The objective of the work should be clearly described at the end of the introduction section

Answer: We thank you for this observation. As you suggested, this paragraph has been moved to the Materials and Methods section. Please see lines 76-87. The objective of the work was mentioned at the end of the introduction section. Please see lines 69-74.

Comment 5. Line 127: it should be indicated that the storage time variable should be analyzed also and the analysis days should be included.

Answer: This was mentioned in the manuscript. Please see lines 79-80. We also added the following sentence: "Samples were taken at intervals of 2 days to evaluate the quality changes of the three Vienna sausage formulations during storage." Please see lines 180-182.

Comment 6. Line 147: Information about illuminant and observer angle should be included.

Answer: We added this information to the manuscript. Please see lines 202-203.

Comment 7. Line 172: PV? Means?

Answer: PV means peroxide value. It was first abbreviated in line 83.

Comment 8. Line 192: The diminution of moisture and the increase in protein content in VSe is quite surprising and it should be better explained. I do not believe is only because of the high substitution…

Answer: We thank you for this observation. The moisture content of the formulation with extract (62.97%) was 1.5% lower than that of the control one (64.40%), which is significant; this is because the mushroom extract, which represents 22.9% of the manufacturing recipe, was used instead of water to prepare the sausages. A possible explanation may be that the extract contributes to the total solids in VSe by containing solubilized substances in the acidic water used to macerate the mushroom powder. We added this information to the manuscript. Please see lines 250-252 in the revised form of it.

Comment 9. The composition of acidic aqueous extract was measured¿? Line 220. Why? What kind of compounds were extracted with the solution apart from polyphenols?

Answer: We did not measure the composition of acidic aqueous extract. At that time, we did not think it could significantly influence the composition of the sausages. However, in support of our findings, Torres-Martínez et al. (2021) found the highest soluble solids concentration in aqueous mushroom extract when they tested the effect of different solvents (water, ethanol, and a mixture of water-ethanol) on the physicochemical properties of edible mushroom extracts. Soluble solids include dissolved sugars, acids, and -at a trace level- vitamins, fructans, proteins, pigments, phenolics, and minerals [Magwaza and Opara, 2015]. Please see lines 252-256 in the revised manuscript.

Comment 10. Line 193: It is not in contrast since you have a decrease of moisture as well as Kanwal et al.

Answer: We thank you for this observation. We corrected it as you suggested. Please see lines 257-259 in the revised manuscript.

Comment 11. Line 200-206: Check the style as a footnote of the table.

Answer: We thank you for this observation. We corrected it as you suggested.

Comment 12. Line 227: Why composition values should change along the storage period and also the energy value? Were the Vienna sausages packaged? This information should be included in the material and methods.

Answer: We thank you for this observation. Vienna sausages were stored unpacked during their shelf life to determine whether the extract, as such or microencapsulated, confers oxidative stability. We added this information to the revised manuscript. Please see lines 179-182.

Comment 13. Line 269: Could there have been microbial spoilage responsible for the pH increase?

We thank you for this observation. We added the explanation in the manuscript. Please see lines 355-357 in the revised manuscript.

Comment 14. Line 408: table 6. Quite surprising that peroxide values and TBARS are higher in samples with encapsulated extract and low polyphenol contents. Could be related to the extraction of polyphenols from the matrix with the encapsulated extracts? A kind of underestimation? I miss a better discussion of the results rather than a description.

Answer: The spectrophotometric method for determining TPC is fast, easy, and cheap; however, it cannot identify polyphenolic compounds individually [López-Fernández et al., 2020]. Therefore, the HPLC method described in subsection 2.4 was further used to estimate them more precisely in Vienna sausages. Please see lines 503-506. That revealed a significantly higher content of some constituents in the enriched formulations than in the control one, thus a higher total content (please see Table 7). However, the PV and TBARS levels were higher in the formulation with microcapsules. This is probably due to the formation of primary and secondary oxidation products in microcapsules during the mushroom extract's spray drying. The high temperature and rapid dehydration during this treatment oxidize, to some extent, the microencapsulation material [Xiong et al., 2021]. Please see lines 529-535.

Another 13 new references were introduced in the revised manuscript.

Round 2

Reviewer 1 Report

Comments and Suggestions for Authors

The comments were clearly and well addressed by authors. The manuscript should be accepted in the present form for publish in this journal.

Reviewer 2 Report

Comments and Suggestions for Authors

accept

Author Response

Dear reviewer,
We thank you very much for your review. Your suggestions were essential and made our paper better.
Best regards,
Authors